Bumblebee size polymorphism and worker response to queen pheromone

Holman Luke luke.holman@anu.edu.au
Centre of Excellence in Biological Interactions, Division of Ecology, Evolution & Genetics, Research School of Biology, Australian National University , Canberra, ACT , Australia
Huber Dezene
Electronic publication date: 2014 Sep 30
Publication date: 2014
Volume: 2
Electronic Location ID: e604
Received 2014 Aug 20; Accepted 2014 Sep 9
Copyright: © 2014 Holman
Copyright year: 2014
Copyright holder: Holman
License: This is an open access article distributed under the terms of the Creative Commons Attribution License, which permits unrestricted use, distribution, reproduction and adaptation in any medium and for any purpose provided that it is properly attributed. For attribution, the original author(s), title, publication source (PeerJ) and either DOI or URL of the article must be cited.
License URL: https://creativecommons.org/licenses/by/4.0/

Keywords: Bombus terrestris, Eusociality, Fertility signal, Reproducible research, Social insects

Funding: Australian Research Council Finnish Academy of Sciences LH received funding from the Australian Research Council (a Laureate Fellowship to H Kokko) and the Finnish Academy of Sciences (Centre of Excellence in Biological Interactions). The funders had no role in study design, data collection and analysis, decision to publish, or preparation of the manuscript.

==============================
Queen pheromones are chemical signals produced by reproductive individuals in social insect colonies. In many species they are key to the maintenance of reproductive division of labor, with workers beginning to reproduce individually once the queen pheromone disappears. Recently, a queen pheromone that negatively affects worker fecundity was discovered in the bumblebee Bombus terrestris, presenting an exciting opportunity for comparisons with analogous queen pheromones in independently-evolved eusocial lineages such as honey bees, ants, wasps and termites. I set out to replicate this discovery and verify its reproducibility. Using blind, controlled experiments, I found that n-pentacosane (C25) does indeed negatively affect worker ovary development. Moreover, the pheromone affects both large and small workers, and applies to workers from large, mature colonies as well as young colonies. Given that C25 is readily available and that bumblebees are popular study organisms, I hope that this replication will encourage other researchers to tackle the many research questions enabled by the discovery of a queen pheromone.

Introduction

Queen pheromones are chemical signals produced by queens (or other fertile females) to communicate with conspecifics, typically other colony members such as workers. They have been implicated in a great many worker responses to queens, including a negative effect on worker ovary development or reproduction (Hoover et al., 2003; Matsuura et al., 2010; Holman et al., 2010; Van Oystaeyen et al., 2014), aggregation around the queen (Keeling et al., 2003), differential behavioral development (Hoover et al., 2003; Vergoz, Schreurs & Mercer, 2007; Matsuura et al., 2010; Holman et al., 2010; Van Oystaeyen et al., 2014), and a wealth of upstream genetic and physiological effects (Kaatz, Hildebrandt & Engels, 1992; Keeling et al., 2003; Grozinger et al., 2003; Malka et al., 2014). Until recently, experimental evidence linking specific queen-produced chemicals to worker responses was largely limited to the well-studied honeybee (Apis mellifera), though there is a wealth of non-experimental evidence that queen pheromones are widespread (reviewed in Kocher & Grozinger, 2011; Van Oystaeyen et al., 2014). In the last four years, additional queen pheromones were experimentally isolated in a few ant species (Smith, Hölldobler & Liebig, 2009; Holman et al., 2010; Smith et al., 2012; Holman, Lanfear & d’Ettorre, 2013; Van Oystaeyen et al., 2014), a wasp and a bumblebee (Van Oystaeyen et al., 2014), and a termite (Matsuura et al., 2010; Matsuura, 2012). It is thus an exciting time to study queen pheromones, since their discovery in these diverse, independently evolved eusocial lineages poses many new questions.

As well as pressing on with queen pheromone research, e.g., by unraveling their full diversity, documenting their effects, and determining their evolutionary significance, I believe it is worthwhile to replicate and validate initial reports of newly discovered pheromones. Replication of empirical work remains somewhat rare throughout the life sciences, likely because novel results are perceived as more valuable (Ioannidis, 2005; Kelly, 2006). Experiments can generate false positives for many reasons, and the false positive rate is probably substantially higher than the 5% implied by the familiar p < 0.05 threshold due to the combined influence of many forms of experimental and statistical bias (Ioannidis, 2005; Simonsohn, Nelson & Simmons, 2014), making replication essential to scientific progress.

I set out to replicate a recent finding that the cuticular hydrocarbon n-pentacosane (hereafter C25) is a queen pheromone that causes sterility in workers of the bumblebee Bombus terrestris. Van Oystaeyen et al. (2014) exposed queenless bumblebee workers to this hydrocarbon or one of four different esters (these five compounds were previously found to be characteristic of queens and fertile workers; Sramkova et al., 2008), then dissected them to determine whether the workers’ ovaries had remained comparatively inactive relative to those of control workers (which were expected to develop their ovaries after the removal of their queen). The four esters had no effect, but C25 significantly reduced worker ovary development relative to the control. Although I was an author on this study, I did not participate in the design or execution of the bumblebee experiment, and I deliberately remained ignorant of the methods used beyond what is written in the paper. Thus, a replication by me is likely to be as similar to the original study as a replication by any other researcher. Due to practical constraints, I opted to test only whether C25 affects worker ovaries, and not to re-test whether the four esters have no effect.

Repeating the experiment also provided the opportunity to gather additional data and try revised methods. In order to test whether previous findings generalize to different colony stages, I used large (c. 300 workers), mature bumblebee colonies, unlike Van Oystaeyen et al. (2014), who used small, developing colonies containing about 20 workers each. Testing the effects of queen pheromones in mature colonies is interesting, because large bumblebee colonies reach the so-called “competition point”, at which many workers begin to reproduce, often while the queen is still present (Van Doorn & Heringa, 1986; Duchateau & Velthuis, 1989; Alaux, Jaisson & Hefetz, 2004). This implies that the response of workers to queen pheromone may decline following the competition point. Additionally, bumblebee colonies naturally produce workers which vary as much as 10-fold in body size, likely because different sized workers are better at different tasks (reviewed in Couvillon et al., 2010), which affects their fecundity (Foster et al., 2004). By recording body size in my experiment, I was able to test whether the effects of C25 differ for large (highly fecund) and small (weakly fecund) workers. Next, I did not use the same ovary scoring system as Van Oystaeyen et al. (2014), since I found it difficult to use during preliminary trials (in particular, I could not reliably identify “regressed” ovaries, if any were indeed present in my sample). Instead, I simply counted developing oocytes. Finally, I used a lower dose of pheromone. Van Oystaeyen et al. (2014) applied 467 µg of C25 per day to each colony, which they estimated to be approximately twice the amount present on the cuticle of a mature B. terrestris queen. To test whether C25 is also efficacious at lower doses (which might more closely mimic the traces of C25 deposited by the queen on the nest substrate as she moves around, though this remains to be measured), I arbitrarily selected a dose of 2 µg per day, or approximately 1/100th of a queen equivalent. In sum, my study is a “partial replication” (reviewed in Kelly, 2006), since it replicates the majority of the design of the original experiment, but studies a later colony stage, adds a measure of worker size, uses a different dose of queen pheromone and an alternative measure of worker ovarian development.

Figure 1 Ovaries by treatment and size.

Treatment with n-pentacosane (C25) reduced the number of oocytes in the ovaries of queenless workers, and there was a strong positive effect of worker body size (note different y axes). The effect of C25 did not differ significantly between Large, Medium and Small workers. The violin plots show the kernel density estimate (i.e., estimated frequency: wide areas contain more data), and the scatter plot shows the raw data. The x coordinates of the raw data are arbitrary: the points were randomly “jittered” horizontally to make overlapping points visible.

Methods

Ten queenright bumblebee colonies (worker number: 300 ± 20) were obtained from Borregaard BioPlant (Denmark), and kept in the plastic cages provided by that company. These colonies had presumably passed the competition point (or would soon do), since they had many more workers than colonies in which worker egg laying has been observed previously (e.g., Van Doorn & Heringa, 1986). The colonies had constant access to sugar water via a feeder and were given pollen ad libitum (the sugar feeder and pollen were obtained from Borregaard BioPlant), and were kept at room temperature. Feeding and pheromone application was performed under red light.

I first removed the queens, using red light illumination to sort through the colonies without anesthesia. Half of the colonies were randomly assigned to the C25 (286931, Sigma-Aldrich) treatment (0.01 µg µl-1 solution in hexane), and half to the hexane-only control (HPLC-grade hexane was used throughout; 34859, Sigma-Aldrich). Every 24 h for 14 days, I added a total of 200 µl of hydrocarbon solution (C25 or hexane; i.e., 2 µg of C25 in the C25 treatment) to the colony by pipetting it through the cage lids onto multiple areas of the cotton wool that lined the nests, taking care to avoid the bees.

After 14 days, colonies were freeze-killed. I then dissected a randomly selected sample of 50–51 workers per colony to determine ovary development. This was accomplished by counting the number of developing oocytes present in the ovaries. Prior to dissection, I scored workers as “Small”, “Medium”, or “Large”, based on whether I estimated them to be in the lower, middle, or upper third of the size range for B. terrestris workers. Dissections and size classifications were performed blind to treatment, and workers were processed in small batches (c. 15) taken from a randomly chosen colony to prevent order effects biasing the results. Since size classification was performed prior to dissection, it was blind with respect to ovary status.

The oocyte count data contained many zeros (394/502 workers had no oocytes in their ovaries), precluding the use of Poisson-based models. I therefore analysed the data with a generalized linear mixed model (GLMM) with negative binomial errors and colony as a random factor (using the function glmer.nb in the lme4 package for R).

Results

C25-treated workers had fewer oocytes in their ovaries than controls (Fig. 1; GLMM: z = −3.04, p = 0.0024, n = 502), and larger workers had more oocytes (Medium vs Large workers: z = −3.30, p = 0.0010; Medium vs Small workers: z = 3.63, p = 0.00029). There was no evidence that workers of different sizes responded differentially to C25 treatment (likelihood ratio test of models with and without the Treatment × Size interaction: p = 0.75; ΔAIC = 3.4; the interaction term was removed when estimating the main effects in the above statistics). The random effect “colony” explained very little variation in oocyte number (variance associated with colony: 1.4 × 10-10, residual variance: 0.55), though I left the colony effect in the model in order to be conservative (the results of a negative binomial generalized linear model lacking colony were near-identical).

The negative effect of C25 treatment on oocytes number might theoretically be explained by a higher frequency of larger-bodied (i.e., more fertile) workers being sampled in the hexane-treated colonies due to chance. In fact, the frequency of “Large” workers was non-significantly higher in the C25 treatment than in the hexane treatment (91 vs 65; binomial GLMM with colony as a random factor: z = 1.68, p = 0.093, n = 502). The sampled workers in the hexane-treated colonies were “Medium”-sized more often than in the C25-treated colonies (126 vs 105), but this difference was not significant (z = 1.17, p = 0.24). Thus, there was no evidence that a chance overabundance of larger-bodied workers in the hexane treatment might explain the observed negative effect of C25 on worker oocyte number.

Discussion

The study replicated Van Oystaeyen et al. (2014)’s finding that C25 is a queen pheromone that negatively affects ovarian development in B. terrestris workers. Furthermore, I found that although larger workers were more fecund, the effect of the pheromone appeared to be consistent across the range of worker sizes.

Because Van Oystaeyen et al. (2014)’s experiment used young bumblebee colonies with few workers while the present study used larger, older colonies, the present results also provide some evidence that the effect of queen pheromone is constant across colony life stages. In turn, this implies that the en masse development of worker ovaries which occurs after the “competition point” cannot be explained by a loss of sensitivity to queen pheromone (since bees from both young and old colonies appear sensitive to the pheromone). Alternative explanations for the onset of worker reproduction (e.g., Van Doorn & Heringa, 1986), such as a reduction in the quantity of pheromone produced by the ageing queen, the declining frequency of queen contact per worker as the colony grows, or the involvement of other signals and cues (e.g., an additional chemical signal, or simply the frequency of worker–worker contacts), thus seem more likely given the present results.

Previous work on bumblebees has suggested that workers signal their fecundity and position in the colony’s dominance hierarchy via their chemical profile, following the competition point. The egg-laying “elite” workers are highly active and aggressive towards other workers (Van Doorn & Heringa, 1986) and are hypothesized to advertise their high fecundity to their nestmates, inducing them to remain sterile or face reprisals in the form of aggression or destruction of their eggs (Amsalem et al., 2009). Given this evidence, and the fact that a constant dose of pheromone might represent a proportionately higher dose for small individuals, one might have expected the pheromone to have a stronger effect on smaller workers. However, I found no evidence that the effect of C25 on ovarian development is size dependent.

Given the consistency of the result that C25 is involved in regulating reproductive division of labor in B. terrestris (namely two independent, blind experiments), it is my hope that other research groups will begin experimenting with this system. Unlike some of the other recently-discovered queen pheromones (e.g., 3-MeC31 in ants and n-butyl-n-butyrate and 2-methyl-1-butanol in termites; Matsuura et al., 2010; Holman, Lanfear & d’Ettorre, 2013), this chemical does not need to be synthesized to order: it can be easily purchased, making it just as readily available as the well-studied honey bee queen pheromone. Though currently unstudied, it is possible that C25 is as multi-functional as other queen pheromones (Le Conte & Hefetz, 2008; Holman, 2010; Kocher & Grozinger, 2011; Matsuura, 2012), and the proximate mechanisms by which it affects worker ovarian development have yet to be discovered.

Supplemental Information

Supplemental Information 1 Raw data from the queen pheromone experiment

Data file containing oocyte counts and size categorisations for individual bees, grouped by treatment and colony of origin (.csv format).

Click here for additional data file.

I am grateful to the Centre for Social Evolution, University of Copenhagen, for hosting these experiments.

Additional Information and Declarations

Competing Interests

Author Contributions

The author declares there are no competing interests.

Luke Holman conceived and designed the experiments, performed the experiments, analyzed the data, contributed reagents/materials/analysis tools, wrote the paper, prepared figure, reviewed drafts of the paper.

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
