# Peer review of "Bumblebee size polymorphism and worker response to queen pheromone"

_PeerJ, doi:10.7717/peerj.604_

## Round 0.1 · original submission · Minor Revisions

Thank you for submitting your MS to PeerJ. I have received both reviews, and both reviewers have suggested minor revisions. Having read the paper myself, I agree with their assessments and I believe that the author can respond adequately to these comments and suggestions.

Reviewer 1 ·

Basic reporting

All fine, perhaps the effect of worker size on oocyte production and pheromone effects in other social insects could be briefly reviewed.

Experimental design

Just two points that need more information:

The introduction and discussion seem to imply that the colonies used here already passed the competition point. If that was so, it should be clearly stated and also be explained how that was verified.

As comparing worker castes is a focus of the study, I think the exact meaning of the worker sizes should be discussed in more detail. The author defines them as percentiles of the colony. This is fine, but I think it is important to know how much size overlap might have been there between classes ACROSS colonies, since this will influence how reliable a negative result such as “worker size has no impact on queen pheromone effect” is. By the way, what determines worker size – is it related to something like colony age?

Validity of the findings

Ok, but see comments on the experimental design

Additional comments

This is a well-written paper reporting a study on the effect of the bumblebee queen pheromone on worker oocyte production. The results confirm findings of an earlier paper and add new information on the effect on workers of different sizes. Also, the paper shows that the pheromone is not only effective small colonies, but in mature colonies as well. I think this study deserves to be published, and I have only two “major” comments that can surely be addressed without problems.

The introduction and discussion seem to imply that the colonies used here already passed the competition point. If that was so, it should be clearly stated and also be explained how that was verified.

As comparing worker castes is a focus of the study, I think the exact meaning of the worker sizes should be discussed in more detail. The author defines them as percentiles of the colony. This is fine, but I think it is important to know how much size overlap might have been there between classes ACROSS colonies, since this will influence how reliable a negative result such as “worker size has no impact on queen pheromone effect” is. By the way, what determines worker size – is it related to something like colony age?

I'd also spend fewer words justifying why experiments should be independently repeated. This is true and deserves a few lines, but it's not the paper's main point. Focus on what's new instead and document that in more detail.

Details:

l 59-73 So much space spent on why replicates are important – can't this be shortened?

l 90 how many workers per colony?

L 93-96 are there any papers on the change of queen chcs around the competition point? Did colonies in your study already reach the competition point?

L 113 I think it's “Aldrich”

l 113: how does the pheromone dose used here relate to the natural concentration in a queenright colony, or to the amount found on a single queen?

L 123-125 are there any objective measures of worker sizes, at least ranges? Were all large workers of all colonies larger than medium workers from all colonies? I guess my question is how stable size classes were across colonies. Although the size classes differed in oocyte numbers, you did not find an influence of size on the pheromone effect. Could any potential effect be obscured by size classes being not clearly defined?

L 149-151 wouldn't that have been accounted for by the glmm anyway?

Results: an estimate of variation due to colony differences might be interesting

Figure: As kernel density plots aren't that common in behavioural biology, I think the figure needs more explanation. In particular, people will be confused as to why the x-axis is random.

L 155-156 Are are there other studies on pheromone effects according to size/caste, perhaps in other social insects? I know there is data on ovary development depending on size, e.g. Dijkstra & Boomsma 2008, Oikos 117: 1892-1906

·

Basic reporting

No Comments

Experimental design

No Comments

Validity of the findings

No Comments

Additional comments

This article fulfills the PeerJ requirements for “scientific and methodological soundness”. The question addressed is important, and the methods are relatively simple but appropriate for the hypotheses of interest. Use of a range of concentrations would have been preferable to generate a dose-response, but I appreciate that this comes with logistical constraints. I agree with the author’s contention that repetition of experiments is important for validation, and yet the paper does have novel aspects to it (e.g. effect of C25 on size). I have only minor comments and suggestions below.

Specific comments
1. L113. What is the biological relevance or rationale for the treatment concentration of 0.01 ug/ul?
2. L113. Is the Sigma chemical coding in reference to the hexane or C25? The code is different from that for hexane on L114. i.e. clarify that the same kind/grade of hexane was used in both treatments.
3. L115. What kind of pollen? Source of pollen?
4. L118. Are we to assume that exposure to C25/hexane was through contact with the contact? Add statement to clarify.
5. L123-127. Why did you score size of bees? It is highly preferable to use continuous data since rank data are so subjective, and this would have been easily done by weighing the bees on a balance. Also, when ranking of individual bees was done, did you randomize within and across colonies to counter changes in your scoring over time?
6. L139-141. I see nothing in Fig 1 to suggest that C25 had any effect on the frequency of workers with oocytes; the raw data and violin plots appear to be evenly distributed on either side of the x-axis (‘density’, which I assume is the same as frequency/proportion) in both treatments. The last part of this sentence simply repeats what was said earlier and can be removed.
7. L160. Change “constant across colony life stages” to “constant among different sized workers”.
8. L160-162. I don’t understand how the results of the current study relate to this statement.
9. L176: Change to “…I found no…”
10. Fig 1. The x-axis title is ambiguous. Would “sample frequency” or “proportion of workers” be better?

---

## Round 0.2 · accepted · Accept

Thank you for your rebuttal and revised MS. I have read your rebuttal and revised MS and you have addressed all of the minor revisions specified by the two reviewers. The revised MS is now acceptable for publication following copyediting, etc.

Please consider making the review history of this MS public, as doing that serves to add value to the publication in general.